# Property Analysis of Double-Sided Composite Waterproofing Sheet for Simultaneous Application on Asphalt Concrete and Latex-Modified Concrete Pavements for Bridge Decks

Hyojin Kang [1], Kyuhwan Oh [2], Kiwon Ahn [2], Bo Jiang [3], Xingyang He [3] and Sangkeun Oh [3,4,*]

1  Department of Architecture of Graduate School, Seoul National University of Science and Technology, 232 Gongneung-ro, Nowon-gu, Seoul 01811, Korea
2  Researcher, Construction Technology Research Institute, Seoul National University of Science and Technology, 232 Gongneung-ro, Nowon-gu, Seoul 01811, Korea
3  School of Civil Engineering and Environment, Hubei University of Technology, No. 28, Nanli Road, Wuhan 430068, China
4  School of Architecture, Seoul National University of Science & Technology, 232 Gongneung-ro, Nowon-gu, Seoul 01811, Korea
*  Correspondence: ohsang@seoultech.ac.kr

**Abstract:** Waterproofing in pavements can determine the waterproofing performance of the entire bridge structure. In this study, two types of pavement layers, asphalt concrete (APC) and latex-modified concrete (LMC), are investigated as options to improve the waterproofing performance of bridge structures with either APC or LMC-type pavement by installing a double-sided adhesive waterproofing sheet. The material performance of the proposed waterproofing sheet was evaluated for deterioration factors such as temperature change, chemical erosion, cracking behavior, and water pressure as stipulated in the Korean industrial standards (hereinafter referred to as KS) for bridge waterproofing materials. The waterproof sheet was directly installed on to specimens with the respective two pavement types to evaluate the field application performance. As a result of the evaluation, the physical waterproofing performance of the proposed waterproofing sheet satisfies all standard quality conditions, and as a result of direct application to APC and LMC pavement, the waterproof performance is at least 8% to 130% higher than the standard quality standard in APC pavement, and LMC pavement shows high performance, up to about 320%. Therefore, it is expected that the newly proposed waterproofing sheet as a bridge deck surface waterproofing material can be considered as a feasible option to improve the waterproofing performance for both APC and LMC pavement.

**Keywords:** latex-modified concrete pavement; asphalt concrete pavement; composite waterproofing sheet; waterproofing performance

## 1. Introduction

Bridge deck cracks and subsequent leakage are always a problem for infrastructure such as long-span bridges and overpasses. Bridges are long-span structures, and cracks on deck are prone to occur due to deflection by continuous traffic load and vibration behavior, rainwater, and other water seepage into cracks, which leads to eroding reinforcing bars and concrete [1,2] (Refer to Figure 1 below for an illustration of water-leakage on bridge deck). This shortens the durability life of the bridge and can be a cause of long-term degradation and, in severe cases, collapse. Therefore, it is essential to apply waterproofing technology to secure and maintain the long-term durability and safety of bridge structures, and as the application of sustainable high-performance waterproofing technology is required, the advancement and employment of waterproofing design standards is emphasized.

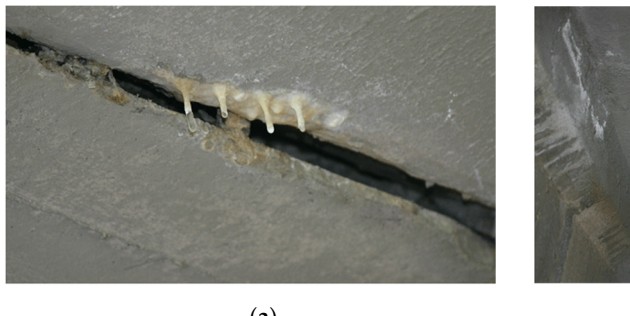
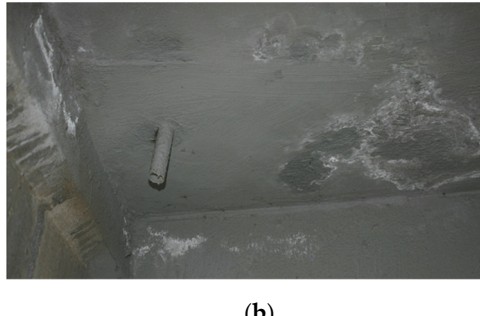

(**a**)                                                                                                (**b**)

**Figure 1.** Types of water-leakage on bridge deck; (**a**) joint leakage, (**b**) slab leakage.

A review of current bridge waterproofing designs and studies indicates that waterproofing methods to protect bridge decks are susceptible to change depending on the type of pavement technology [3,4]. Haynes et al. provide a study on the tendencies of failure adhesion of commonly used asphalt layer waterproofing to the pavement overlay, and provide an outline on effective waterproofing methods and strategies [5]. Russell, H. published an extensive report on the various waterproofing material types and methods, but among them, new bridges sometime do not employ the usage of waterproofing layers [6]. Liu Xueyan et al. researched and modelled the adhesion performances of two different types of concrete pavement of bridge decks with asphalt overlay, and outline the necessary parameters for accurate analysis [7]. Robert J. Frosch discusses the possibility of using thin overlays that protect the pavement layer, a newly proposed technology that is intended to supplement the common adhesion problems between asphalt overlap and concrete [8].

In some cases, the presence or absence of a waterproofing layer is determined by the type of paving methods. Most bridge deck slabs (bridge pavement structures) are comprised of high strength and rigid concrete, and then asphalt concrete (APC, hereinafter referred to as APC) or latex-modified concrete (LMC, hereinafter referred to as LMC) is subsequently installed as a pavement layer (or material).

In the case of APC pavement as a bridge waterproofing method, chloroprene-rubber-based coating or modified asphalt sheet has been conventionally used for a long time [5,6]. This is because when APC is mixed in high-temperature conditions (temperature can increase up to 180 °C) [7] with asphalt and aggregate (gravel) as pavement materials, the chloroprene rubber or modified asphalt sheet adheres well to APC and can withstand the high temperature and rolling pressure of APC. Chloroprene rubber coating or modified asphalt sheets have produced successful results with waterproofing to protect the leakage of bridge decks with continuous performance maintenance in Korea [7], but it recent discoveries indicated that chloroprene rubber coating is a cause of air pollution [8,9], modified asphalt sheets are mainly used instead in the Korean market [10,11]. Modified asphalt sheets, in comparison to asphalt overlays, have higher adhesion performance, but are susceptible to wear and damage due to shear force, which could potentially lead to adhesion failure.

LMC pavement is also widely used as a bridge and road pavement material in addition to APC [12]. LMC is a paving material made by adding a latex-based polymer into concrete [13]. This is a technology that provides elasticity and watertightness to concrete to reduce cracking, and waterproofing performance to the concrete itself and is often considered to not require a waterproofing layer on the bridge deck surface [14]. Traditionally, concrete pavements are subject to a higher rate of thermal expansion and deflection, as they are directly exposed to wheel load and thermal variation from environmental sources. Concrete pavements are, therefore, known to be susceptible to cracks, freeze–thaw damage, etc., occurring in the pavement layer and lose waterproofing performance (Refer to Figure 2 for cases of deterioration in bridges). Despite their design to be inherently watertight, the problem of leakage of the bridge deck surface persists in concrete-based pavements, including LMC [15]. In this regard, in the case of LMC pavement, the necessity of providing

a waterproof layer is emphasized more. In fact, it is a big problem that there are currently no waterproofing materials with the performance that can be attached to the LMC pavement layer [16]. The general waterproofing materials that currently exist do not directly adhere to the hydration water contained in the LMC. Therefore, a bridge designed with an LMC pavement layer cannot be constructed with a waterproofing layer, due to its structural characteristics of adhesion problems on the LMC pavement layer, waterproofing layer, and bridge deck surface [17,18].

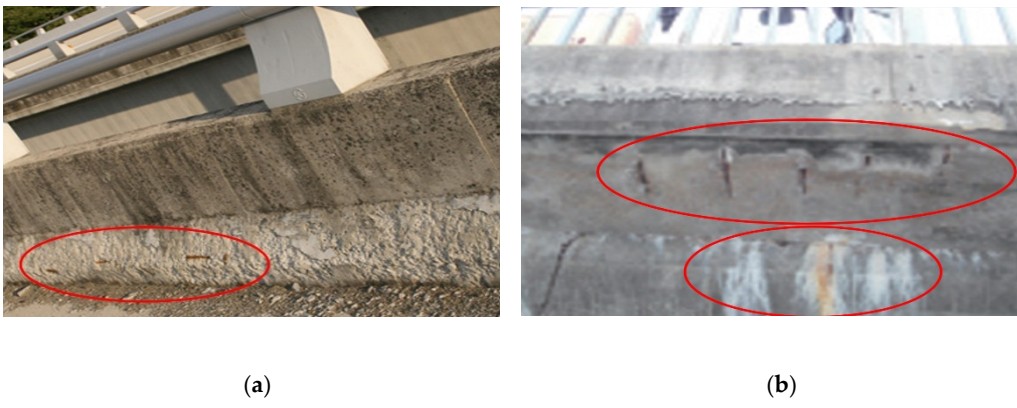

(**a**)                                    (**b**)

**Figure 2.** Deterioration factors of waterproofing materials for bridges and deterioration of structures; (**a**) corrosion of concrete rebar due to leakage, (**b**) deterioration of bridge structures due to influent water.

In this current situation, the absence of a waterproofing material with watertight adhesion to the LMC pavement layer may also be a problem, but the application of materials with different performance depending on the pavement layer material may cause another problem [19,20]. Therefore, it is necessary to develop waterproofing materials suitable for both the material components and construction conditions of APC and LMC.

By improving the existing modified asphalt sheets used for bridge deck waterproofing, a composite waterproofing sheet that can be applied to both APC and LMC was developed. One side of this composite waterproofing sheet can be blind-side applied to LMC by a hydration reaction with water, and the other side can withstand the high temperature and rolling pressure of the APC.

In this study, the adhesion characteristics of the proposed double-sided bonding waterproofing layer on LMC and APC pavement specimens were investigated after high temperature and shear pressure treatment, respectively, were applied. Through the evaluation, it was intended to suggest the possibility of using a new waterproofing material for bridge deck surfaces in the future.

## 2. Properties and Test Methods of Newly Proposed Waterproofing Sheet and Layer

### 2.1. Properties of Newly Proposed Composite Waterproofing Sheet

2.1.1. A Double-Side Bonding Waterproofing Sheet

The newly proposed double-side bonding waterproofing sheet investigated in this study is illustrated in Figure 3. This material was developed based on a modified asphalt waterproofing sheet impregnated with an asphalt compound impregnated in a layer reinforced with a non-woven fabric, and the modified acrylic emulsion was applied to the other side to secure adhesion through hydration reaction when placed in contact with the cast concrete.

This proposed waterproofing sheet is a double-side bonding type comprised of a modified asphalt sheet that can withstand the high-temperature conditions that occur during APC pavement construction, and can be installed by hydration reaction with latex acrylic emulsion that occurs during LMC pavement construction. The intended purpose of this material is to provide an option for concrete-(LMC) or asphalt (APC)-based pavements to use waterproofing sheets with stable adhesion when waterproofing is highly

recommended. Figure 4 shows the illustrated concept of the proposal for improving the waterproofing performance of both APC and LMC pavements.

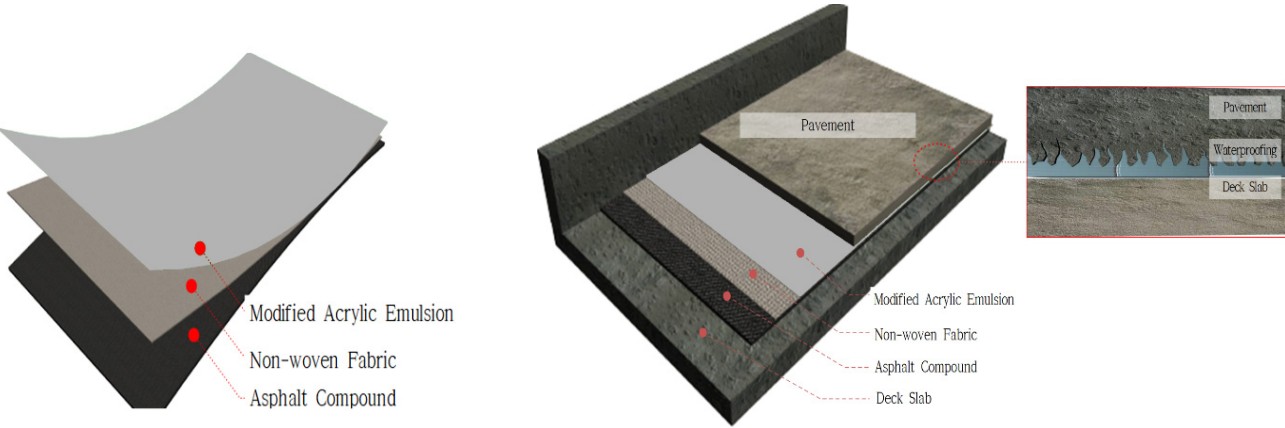

**Figure 3.** APC and LMC pavement structure with a double-side bonding waterproofing sheet.

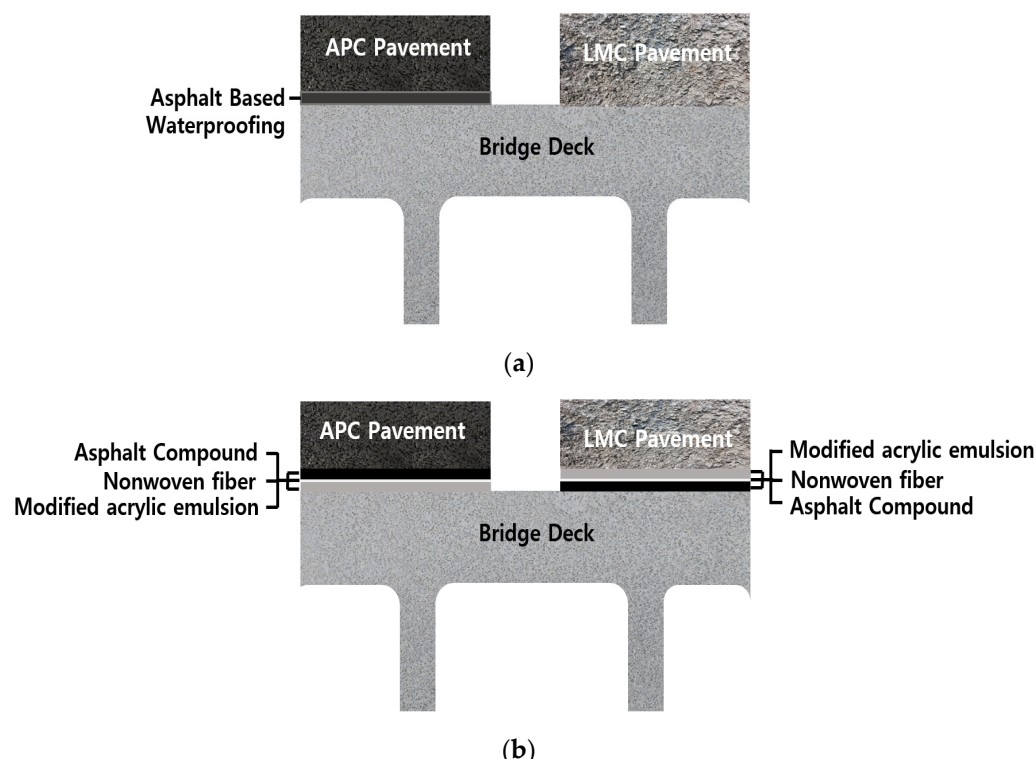

**Figure 4.** Proposed research scope for improving waterproofing performance of bridge pavements illustrated; (**a**) conventional design of APC and LMC pavement comparison, (**b**) proposed waterproofing method using the double-sided bonding waterproofing sheet concept for both APC and LMC pavements.

### 2.1.2. Properties of Modified Latex Acrylic Emulsion

The upper part of the double-side bonding waterproofing sheet induces a hydration reaction with the poured concrete through the modified latex acrylic emulsion. To determine the appropriate mixing ratio based on weight percentage of the LMC constituent materials, five mixing ratios were selected, and the performance of the waterproof layer formation was evaluated and subsequently selected based on highest adhesion strength. Through this precursory test, adhesion according to the acrylic emulsion mixing condition was reviewed to verify satisfactory watertight adhesion in accordance with standard testing, and the

theoretical mixing formulation ratio 2 was set as the standard mixture for the purposes of this study. The mixing additives were adjusted as shown in Table 1.

**Table 1.** Adhesive performance assessment based on different component mixture ratios of modified latex acrylic emulsion.

| Components | Mixture Ratio Based on Weight Percentage (%) | | | | |
| --- | --- | --- | --- | --- | --- |
| | **Ratio 1** | **Ratio 2** | **Ratio 3** | **Ratio 4** | **Ratio 5** |
| Functional cement | 10 | 15 | 18 | 23 | 25 |
| Silica fine powder | 25 | 38 | 32 | 35 | 48 |
| New synthetic resin | 85 | 41 | 38 | 55 | 46 |
| Tackifier | 4 | 5 | 3 | 0.5 | 1.0 |
| Color pigment | 10 | 6 | 6 | 10 | 5 |
| Antifoam and additives | 3 | 4 | 2.5 | 10 | 12 |
| Adhesive performance confirmation $(N/mm^2)$ | 0 (no adhesion) | 24 | 7.5 | 8.6 | 0 (no adhesion) |

Based on the results in Table 1, the double-sided bonding waterproofing sheet with modified acrylic emulsion layer under the conditions of formulation ratio 2 is found to have the highest adhesion strength performance (24 $N/mm^2$). Based on the results, it was determined that the adhesion strength is dependent on the relative ratio of the additives and the new synthetic resin components of the mixture ratio (limited to the specifications outlined in the test method KS F 4934 "18" "Self Adhesive Rubberized Asphalt Sheet" [18]). The modified asphalt compound layer with the reinforced non-woven fabric layer was subsequently manufactured for physical property evaluation.

2.1.3. Required Physical Properties and Evaluation Items of Waterproofing Sheet

This study was conducted in compliance with Korean standard KS F 4931-"17" "Sheet Waterproofing Material for Concrete Bridge Deck" [4] test to verify the physical properties of the double-sided composite waterproof sheet intended to secure waterproof performance for both APC and LMC pavement layers of the bridge deck. The nine test items are tensile performance (including alkaline components and temperature conditions), permeability, chloride ion penetration resistance, impact resistance, heat (150 °C) resistance stability, low temperature flexibility, overlap section tensile strength, fatigue resistance, and crack resistance, as shown in Table 2 below (methodology of the parameters outlined in KS F 4931 (Table 2) are in compliance with similar methods found in the American Society of Testing Methods (ASTM) as KS F 4931 was drafted using ASTM and Japanese standard (JS) as reference).

*2.2. Preparation of Pavement and Deck Concrete Materials*

For this study, conventional material composition for the APC pavement is shown in Table 3 below. Using asphalt compound, coarse aggregate, fine aggregate, and filling material (limestone powder) as a pavement material, the APC pavement for the specimen was prepared. In Table 4, the mixture ratio of LMC pavement layer is shown, and outlines the mixture ratios based on weight percentage for cement-based binders, fine aggregates, coarse aggregates, and water, similar to general concrete, and liquid latex is additionally

added. The base concrete (deck slab) as waterproofing substrate that was used for specimen preparation were based on ordinary Portland cement concrete, as shown in Table 5.

**Table 2.** Physical property evaluation criteria and performance standard of waterproofing sheet.

| Item | Criteria | | Quality Standard |
|---|---|---|---|
| Tensile performance | Tensile strength (N/mm) | Untreated | More than 13.0 |
| | | Alkali-treated | |
| | | Temperature-variation-treated | |
| | Elongation (%) | Untreated | More than 33 |
| | | Alkali-treated | |
| | | Temperature-variation-treated | |
| | Permeability | | Should not leak |
| | Chloride ion penetration resistance (Coulombs) | | Less than 100 |
| | Impact resistance | | Holes should not form |
| | Heat resistance stability (%) | 150 °C, 30 min | Within $\pm$ 2.0 deformation |
| | Low temperature flexibility | −20 °C | No cracks should form |
| | Overlap section strength (N/mm) | | More than 5.0 |
| | Fatigue resistance | | No cracks, tears, or breakage should form |
| | Crack resistance | −20 °C | No cracks, tears, or breakage should form |

Note: Same applies regardless of the asphalt or concrete of the pavement layer according to the material property evaluation item.

**Table 3.** APC pavement material composition.

| Composition | Asphalt | Coarse Aggregate | Fine Aggregate | Filler (Limestone Powder) | Others |
|---|---|---|---|---|---|
| Mixture ratio based on weight percentage (%) | 4~5 | 65~80 | 14~26 | 1~2 | 2~4 |

**Table 4.** LMC pavement material composition.

| Composition | Cement Binder | Fine Aggregate | Coarse Aggregate | Water | Latex | Others |
|---|---|---|---|---|---|---|
| Mixture ratio based on weight percentage (%) | 15~20 | 40~45 | 30~35 | 5~10 | 1~3 | 1~2 |

**Table 5.** Base (substrate as deck slab) concrete mixture condition.

| Mixture Ratio Based on Weight Percentage | | | | | | |
|---|---|---|---|---|---|---|
| Water/Cement Ratio (%) | Unit water (kg/m³) | Small Aggregate (%) | Air Content (%) | Cement (kg/m³) | Sand (kg/m³) | Gravel (kg/m³) |
| 50 | 82.5 | 43 | 4.1 | 165 | 646 | 895 |

*2.3. Test Methods for Performance Evaluation of Waterproofing Layer Applied to Pavement Layer*

2.3.1. Specimen Preparation and Test Methods of Waterproofing Sheet

(1)    Specimen fabrication

The specimens of APC and LMC pavement with the applied waterproofing sheet specimens were fabricated according to the specimen manufacturing method of the Korean standard KS F 4931-"17" [sheet waterproofing materials for concrete bridges], as shown in Figure 5, and placed in the constant room-temperature setting (20 °C ± 3, RH 65%) for curing for 3 days. The 20 × 20 cm-sized double-sided composite waterproof sheets were applied for each specimen type, and thickness of the concrete substrate (bottom part) and the pavement substrate (upper part) were prepared. For each substrate type (APC and LMC), three specimens were prepared.

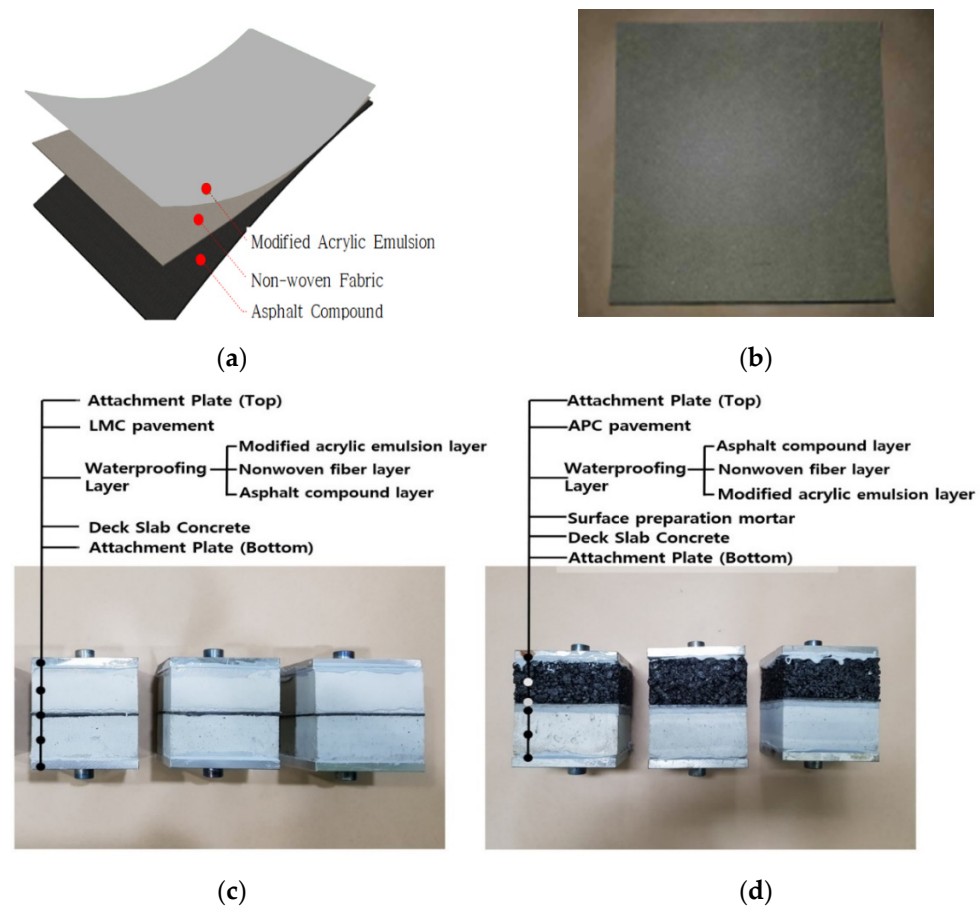

**Figure 5.** APC and LMC pavement specimens; (**a**) a double-sided waterproofing sheet, (**b**) surface of waterproofing sheet, (**c**) LMC pavement specimen with applied waterproofing sheet, (**d**) APC pavement specimen with applied waterproofing sheet.

(2)    Test method

The test methods were conducted by applying the test method of the Korean standard KS F 4931-"17" [sheet waterproofing materials for concrete bridges], as shown in Figures 6 and 7. The performance evaluation items for tests of APC and LMC pavement specimens with applied waterproofing sheet are shear adhesive and deformation, tensile adhesive strength, and water immersion properties, as shown in Table 6.

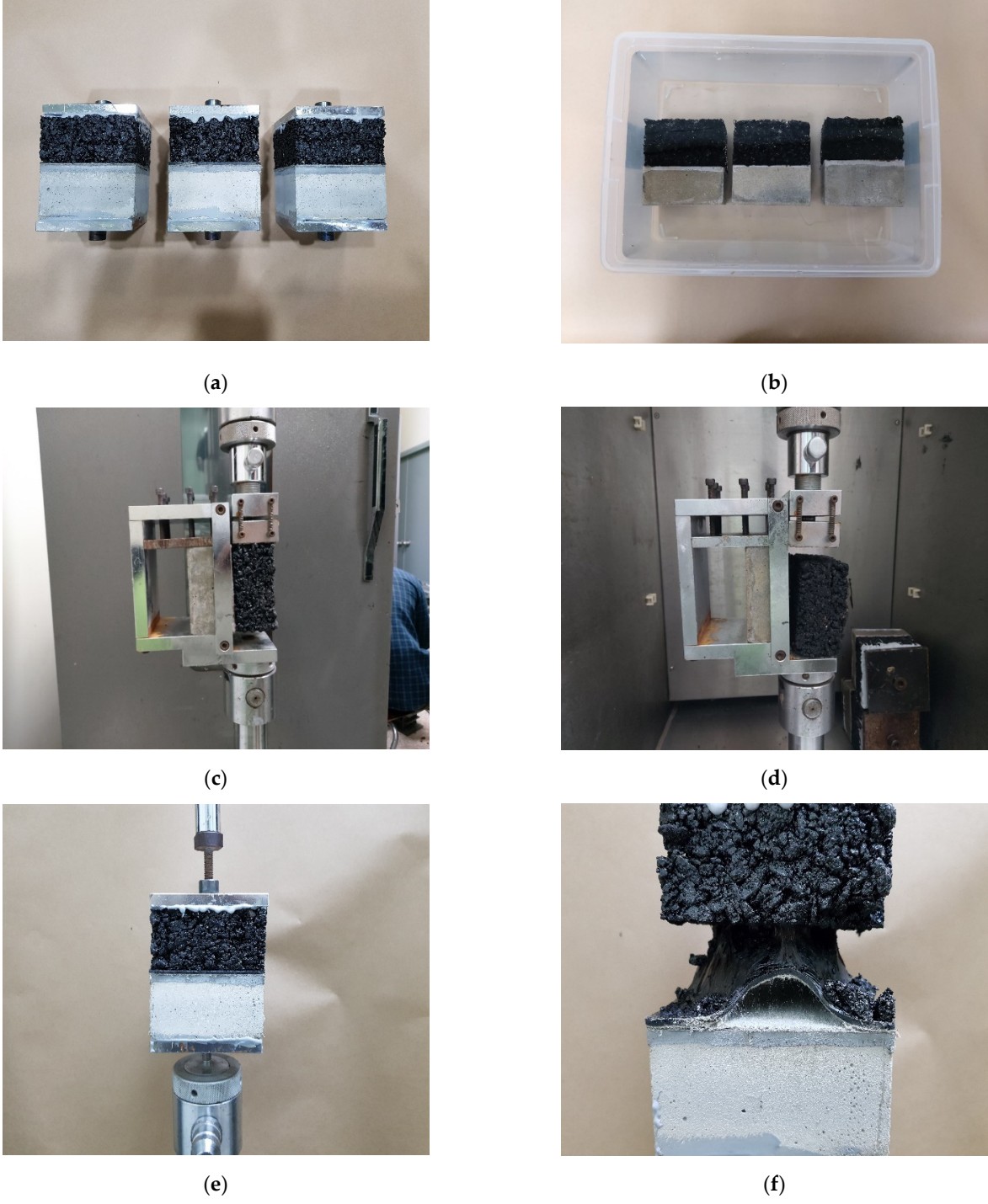

**Figure 6.** Evaluation of methods of APC pavement specimens with applied waterproofing sheet; (**a**) APC specimens, (**b**) water immersion test, (**c**) shear adhesive and deformation test, (**d**) shear and deformation test result, (**e**) tensile test, (**f**) tensile adhesive strength test result.

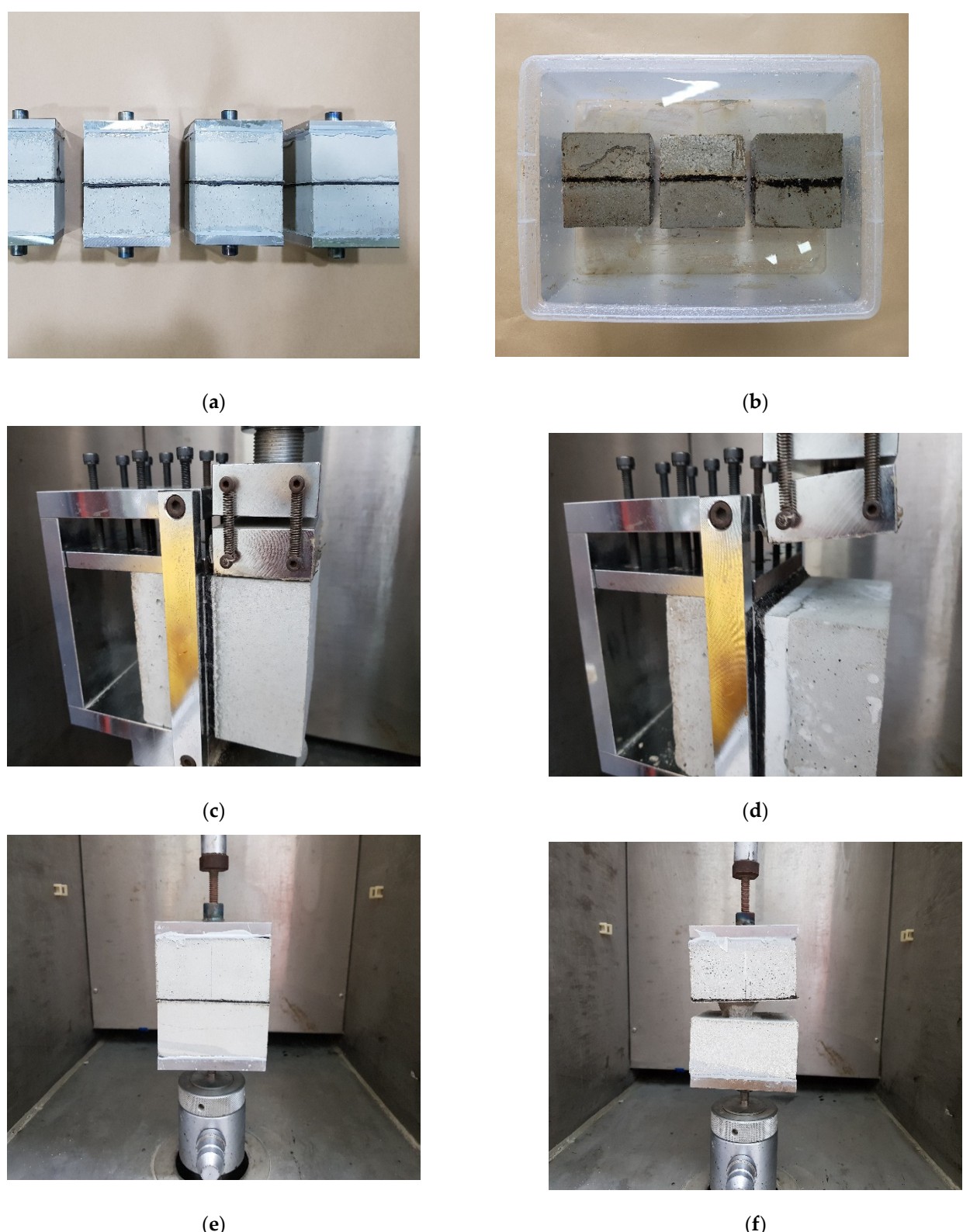

**Figure 7.** Evaluation of methods of LMC pavement specimens with applied waterproofing sheet; (**a**) LMC specimens, (**b**) water immersion test, (**c**) shear and deformation test, (**d**) shear adhesive and deformation test result, (**e**) tensile test, (**f**) tensile adhesive strength test result.

**Table 6.** Evaluation criteria and quality standards of waterproofing sheet.

| Items | | | Quality Standards |
|---|---|---|---|
| Shear adhesive performance | Shear adhesive strength (MPa) | −20 °C | >0.80 |
| | | 20 °C | >0.15 |
| | Shear adhesive deformation (%) | −20 °C | >0.5 |
| | | 20 °C | >1.0 |
| Tensile adhesive strength (MPa) | | −20 °C | >1.2 |
| | | 20 °C | >0.6 |
| Tensile adhesive strength after 7 days of water immersion (%) | | 20 °C | 70% or more before water immersion |

Note: According to the evaluation of the physical properties of the construction method, the upper pavement layer is made of asphalt concrete and concrete differently during the test piece production process, and the test is carried out.

2.3.2. Performance Evaluation Items and Quality Standard of Waterproofing Layer between Pavement and Bridge Deck Concrete

The physical property of the composite structure specimen was evaluated based on Korean standard (KS) testing standards. A test method was conducted to evaluate the applicability of a double-sided bonding waterproofing sheet for bridge deck surface that demonstrates watertight adhesion performance with APC or LMC pavement layers. The double-sided bonding waterproofing sheet was installed onto to the asphalt compound of APC or the hydration reaction of LMC, such that watertight adhesion can be secured regardless of the pavement layer material. The quality of the waterproofing layer (sheet) between pavement and bridge deck concrete ensured the performance above the KS standard quality values. The test items and required quality standard of waterproofing sheet are shear adhesive performance (strength and deformation), tensile adhesive strength (in air and water immersion), and the temperature condition are −20 °C and 20 °C, as shown in Table 6, of KS F 4931-"17" "Sheet waterproofing Material for Concrete Bridge Deck" (methodology of the parameters outlined in KS F 4931-"17" (Table 6), and are in compliance with similar methods found in the American Society of Testing Methods (ASTM), as KS F 4931 was drafted using ASTM and Japanese standard (JS) as reference).

**3. Evaluation Results and Considerations**

*3.1. Performance of Double-Sided Bonding Waterproofing Sheet*

3.1.1. Basic Property Test Results

The basic evaluation results of a double-sided bonding waterproofing sheet are shown in Table 7 below. As a result of the evaluation, it is confirmed that the corresponding performance of the newly proposed waterproofing layer shows improved performance over quality standard values of nine evaluation items of KS F 4931-"17" "Sheet Waterproofing Material for Concrete Bridge Deck." Most of the performance of the proposed double-sided bonding waterproofing sheet more than satisfies the quality standard of KS requirement. In Table 7, under the column 'test results,' arrows (↑) mark the increase in the value in terms of percentage ratio from the minimum requirement outlined in the test standard. The tensile strength value increases by about 182%, elongation rate by about 357%, the chloride ion penetration resistance by 14%, and the high-temperature stability by 45%. Furthermore, there are no other defects found in other parameters including water permeability, impact resistance, fatigue resistance, crack resistance, etc.

**Table 7.** Performance test results of double-sided waterproofing sheet.

| Evaluation Items | | | Test Results | Quality Standard |
|---|---|---|---|---|
| Tensile performance | Tensile strength (N/mm) | Untreated | Length | 23.7 (182.31% ↑) | >13.0 |
| | | | Width | 19.3 (148.46% ↑) | |
| | | Alkali-treated | Length | 24.2 (186.15% ↑) | |
| | | | Width | 19.8 (152.31% ↑) | |
| | | Temperature-variation-treated | Length | 21.1 (162.30% ↑) | |
| | | | Width | 16.8 (129.23% ↑) | |
| | Elongation (%) | Untreated | Length | 59 (178.78% ↑) | >33 |
| | | | Width | 73 (221.21% ↑) | |
| | | Alkali-treated | Length | 118 (357.57% ↑) | |
| | | | Width | 102 (309.09% ↑) | |
| | | Temperature-variation-treated | Length | 107 (324.24% ↑) | |
| | | | Width | 98 (296.96% ↑) | |
| Permeability | | | No leakage | Should not leak |
| Chloride ion penetration resistance (Coulombs) | | | 86 (14% ↑) | <100 |
| Impact resistance | | | No holes formed | Holes should not form |
| Heat resistance stability (%) | 150 °C, 30 min | | Length | 1.1 (45% ↑) | Within ±2.0 deformation |
| | | | Width | 0.9 (55% ↑) | |
| Low temperature flexibility | −20 °C | | No cracks | |
| Overlap section strength (N/mm) | | | 8.5 (170% ↑) | >5.0 |
| Fatigue resistance | | | No cracks, tears, or breakage | No cracks, tears, or breakage should form |
| Crack resistance | −20 °C | | No cracks, tears, or breakage | No cracks, tears, or breakage should form |

### 3.1.2. Consideration

As a result of the proposed waterproofing material property evaluation test, it is found that the waterproofing performance satisfies the required standard criteria when an acrylic emulsion layer is applied to the modified asphalt sheet to form a composite waterproofing layer. Therefore, if the adhesion to LMC and APC pavement layers applied to the proposed waterproofing sheet is verified, this material can be considered as a feasible option for reinforcing the waterproofing performance for bridge deck surfaces.

### 3.2. Performance Properties of Waterproofing Sheet after Construction of APC or LMC Pavement Layer

### 3.2.1. Effect of APC Pavement Layer

(1) Test results

Table 8 shows the performance evaluation results of the waterproofing sheet effect on the APC pavement layer. Under the column 'test results,' arrows (↑) mark the increase in the value in terms of percentage ratio from the minimum requirement outlined in the test standard. The shear adhesive performance (strength and deformation) and tensile adhesive strength values of the waterproof sheet attached to the APC pavement layer are higher than the KS quality standards in all evaluation items under both −20 °C and 20 °C conditions.

**Table 8.** Performance test results of waterproofing sheet effect on APC pavement layer.

| Evaluation Items | | | Test Results | Quality Standard |
|---|---|---|---|---|
| Shear adhesive performance | Shear adhesive strength (MPa) | −20 °C | 0.89 (11.25% ↑) | More than 0.80 |
| | | 20 °C | 0.28 (86.67% ↑) | More than 0.15 |
| | Shear adhesive deformation (%) | −20 °C | 0.8 (60% ↑) | More than 0.5 |
| | | 20 °C | 2.3 (130% ↑) | More than 1.0 |
| Tensile adhesive strength (MPa) | | −20 °C | 1.3 (8.33% ↑) | More than 1.2 |
| | | 20 °C | 0.7 (16.67% ↑) | More than 0.6 |
| Tensile adhesive strength after 7 days of water immersion (%) | | 20 °C | 89% (27.14%↑) | 70% or more before water immersion |

(2)    Consideration

As a result of evaluating the performance of the waterproofing layer after installing the APC pavement layer, shear adhesive strength is about 11% higher at −20 °C and about 86% higher at 20 °C, and shear adhesive deformation is about 60% higher at −20 °C and about 130% higher at 20 °C. The tensile adhesive strength is about 8% higher at −20 °C and about 16% higher at 20 °C, and the tensile adhesive strength after immersion in water is about 27% higher. Further investigation will be required before the proposed double-bonding waterproofing sheet is feasible for providing a stable waterproof performance even when applied to APC pavements.

3.2.2. Effect of LMC Pavement Layer

(1)    Test results

Table 9 shows the performance evaluation results of the waterproofing sheet effect on the LMA pavement layer. Under the column 'test results,' arrows (↑) mark the increase in the value in terms of percentage ratio from the minimum requirement outlined in the test standard. The shear adhesive performance (strength and deformation) and tensile adhesive strength values of the waterproof sheet attached to the LMC pavement layer are higher than the KS quality standards in all evaluation items under both −20 °C and 20 °C conditions.

**Table 9.** Performance test results of waterproofing sheet effect on LMC pavement layer.

| Criteria | | | Test Results | Quality Standard |
|---|---|---|---|---|
| Shear adhesive performance | Shear adhesive strength (MPa) | −20 °C | 0.91 (13.75% ↑) | More than 0.80 |
| | | 20 °C | 0.33 (120% ↑) | More than 0.15 |
| | Shear adhesive deformation (%) | −20 °C | 1.3 (160% ↑) | More than 0.5 |
| | | 20 °C | 4.2 (320% ↑) | More than 1.0 |
| Tensile adhesive strength (MPa) | | −20 °C | 1.3 (8.33% ↑) | More than 1.2 |
| | | 20 °C | 0.8 (33.33% ↑) | More than 0.6 |
| Tensile adhesive strength after 7 days of water immersion (%) | | 20 °C | 94% (28.57% ↑) | 70% or more before water immersion |

(2)    Consideration

As a result of evaluating the performance of the waterproofing material after installing the LMC pavement, shear adhesive strength is found to be about 13% higher at −20 °C and about 120% higher at 20 °C, and shear adhesive deformation is about 160% higher

at −20 °C and about 130% higher at 320 °C. The tensile adhesive strength is about 8% higher at −20 °C and about 33% higher at 20 °C, and the tensile adhesive strength after immersion in water is about 28% higher. Further investigation will be required before the proposed double-bonding waterproofing sheet is feasible for providing a stable waterproof performance even when applied to LMC pavements.

### 3.2.3. Comparison of the Performance of the Waterproofing Layer in the APC and LMC Pavement

The comprehensive comparison results according to three sets of specimens of APC pavement and LMC pavement are based on the results in Sections 3.2.1 and 3.2.2 and are summarized in Figures 8 and 9. When checking the detailed physical properties, the proposed waterproofing layer exhibits better performance in the LMC pavement than in the APC pavement. To summarize the specific details, the shear adhesive strength is about 2% higher at −20 °C and about 17% higher at 20 °C, shear adhesive deformation is about 62% higher at −20 °C and about 82% higher at 20 °C, tensile adhesive strength is almost equal at −20 °C, about 14% higher at 20 °C, and about 1% higher in tensile adhesive strength after immersion in water. Through this performance comparison, the proposed waterproofing sheet has a performance above the KS quality standards, and even when applied to each of the APC and LMC pavements, all the specimens secure a performance above the quality standard, indicating that the practical use of the waterproofing material for the bridge deck surface is feasible and can be considered as an option for use when reinforced waterproofing is required.

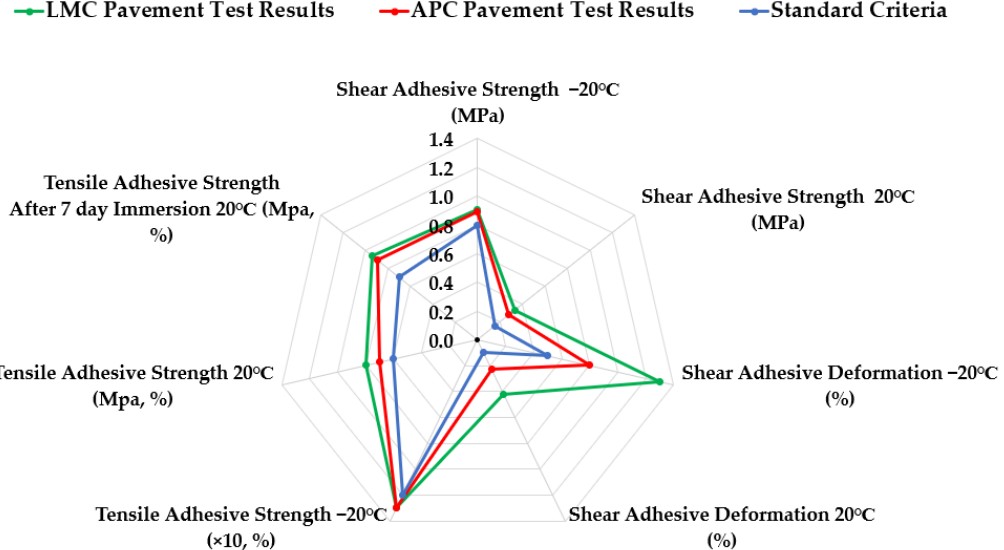

**Figure 8.** Comparison of the performance of the waterproofing layer in the APC and LMC pavements.

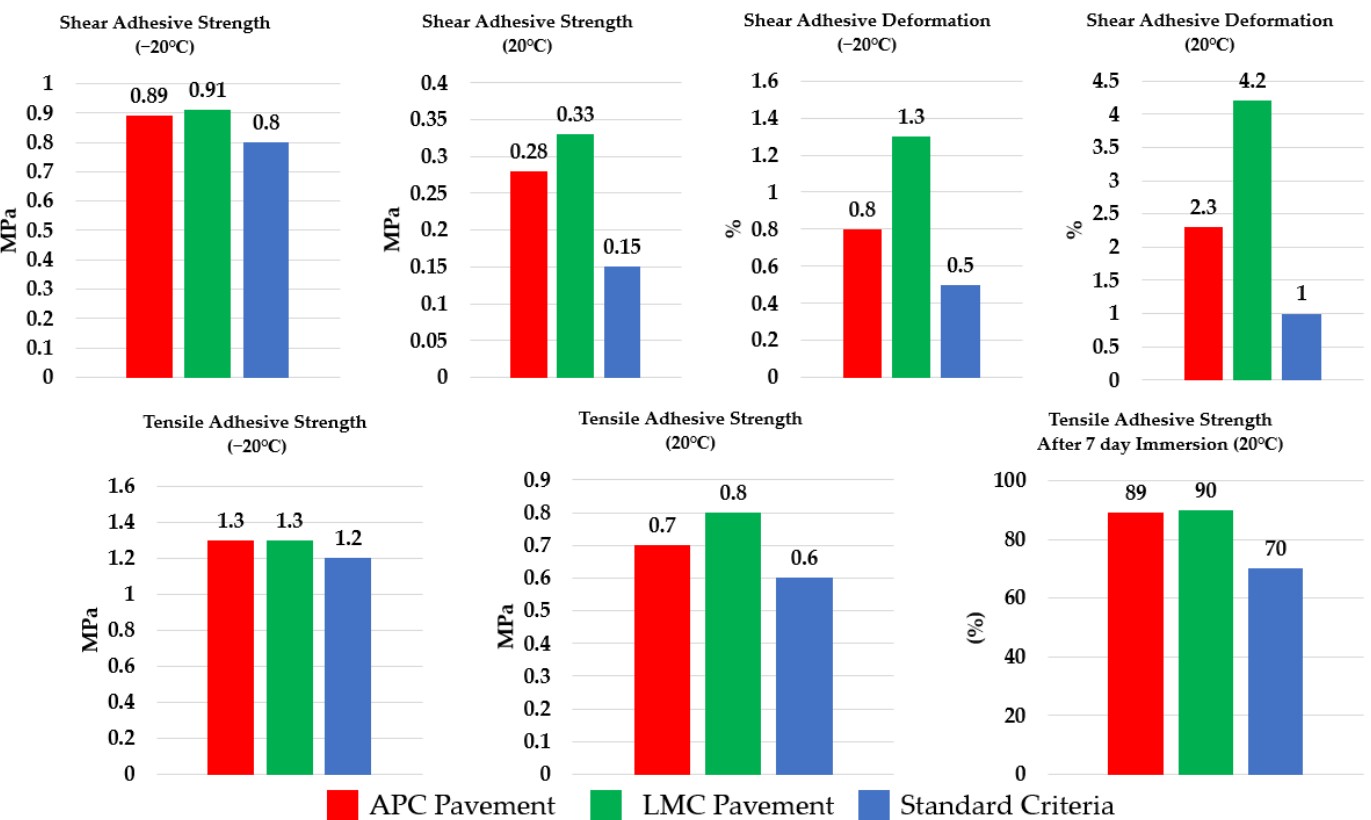

**Figure 9.** Individual comparison of physical property evaluation results of waterproofing sheets in APC and LMC pavement layers.

## 4. Conclusions

This study was conducted from the effort to provide the means to improve the existing waterproofing methods for pavement technologies for bridge deck surfaces. A double-sided bonding composite waterproofing sheet that can be applied to both APC and LMC pavement layers used on the bridge deck was proposed as a new waterproofing material, and its physical properties and applicability were verified in accordance to KS standard test methods. The conclusion of this study is as follows:

(1) For the purpose of preventing water leakage to ensure long-term durability of concrete bridges, a double-sided composite waterproofing sheet manufactured by laminating an asphalt compound layer and a modified acrylic emulsion layer on both sides with a non-woven fabric as the central material was proposed as a new bridge surface waterproofing material. As a result of measuring the resistance performance of deterioration factors such as temperature change, chemical influence, crack behavior, and water pressure for the newly proposed waterproofing sheet, according to the KS test method related to the waterproofing material for bridges, it is confirmed that the performance exceeds the KS quality standard;

(2) As a result of evaluating the applicability of this waterproofing sheet when used with the APC pavement, it is confirmed that the material exceeds the performance requirement outlined in the KS standard by at least 8% and at most 130% in various performance items. Therefore, the waterproofing sheet proposed in this study is expected to exhibit stable waterproof performance even when applied to the APC pavement. In the case of LMC pavement, it is verified that the material exceeds the performance requirements outlined in the KS standard, at most, by about 320% in various performance items;

(3) Comparing the results of using this waterproofing sheet for APC and LMC pavements, the performance is shown to be relatively higher with the LMC pavement, and it is

predicted that this result is due to the difference in the waterproofing layer adhesion caused by the difference in the properties of the waterproofing sheet material and the pavement material.

As it stands, the double-sided bonding waterproofing sheet is verified as feasible for use as an option for waterproofing materials in bridge surfaces, regardless of the type of APC or LMC pavement. However, as experiments conducted in this study are limited to performance evaluation and standard requirements of Korean standards, further investigation is required before this material can be certified as viable for use in the international setting.

**Author Contributions:** Conceptualization, H.K., K.O., K.A., S.O., B.J. and X.H.; methodology, H.K., K.O. and S.O.; experimental plan, H.K., K.A., K.O., S.O., B.J. and X.H.; formal analysis, investigation, resources, data curation, H.K. and K.O.; writing—original draft preparation, H.K., K.O., K.A., S.O., B.J. and X.H.; writing—review and editing, S.O., B.J. and X.H.; supervision, S.O., B.J. and X.H.; project administration. All authors have read and agreed to the published version of the manuscript.

**Funding:** This research was funded by research support program of Seoul National University of Science and Technology in 2021.

**Institutional Review Board Statement:** Not applicable.

**Informed Consent Statement:** Not applicable.

**Data Availability Statement:** Not applicable.

**Conflicts of Interest:** The authors declare no conflict of interest. The founding sponsors had no role in the design of the study; in the collection, analyses, or interpretation of data; in the writing of the manuscript, and in the decision to publish the results.

## Abbreviations

| | |
|---|---|
| LMC | Latex-modified concrete |
| APC | Asphalt concrete |
| KS | Korean standard |

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
