# Peer review of "Property Analysis of Double-Sided Composite Waterproofing Sheet for Simultaneous Application on Asphalt Concrete and Latex-Modified Concrete Pavements for Bridge Decks"

_applsci, doi:10.3390/app12199779_

Round 1

Reviewer 1 Report

The authors proposed a double-sided bonding composite waterproofing sheet as a new waterproofing material that can be simultaneously applied to the APC or LMC pavement layers used on the bridge deck. The methodology was described comprehensively and the results are meaningful. However, there are some problems existing in this paper which the authors must pay attention to deal with.

1.      The abstract is suggested to be condensed.  

2.      The Figure 2 is not necessary.

3.      Please offer the full name for the term which is mentioned at the first time. For example, the KS.

4.      Some expressions in the manuscript are not clear. An English native speaker is suggested to carefully proofread it again. For example, the sentence between 108 and 112 is suggested to be reorganized into 2 shorter sentences.

5.      Some words are too small in Figure 5. It is better to use the same font size in the figures and the texts.

6.      Aspect ratio in some figures seems to be changed, e.g., Figures 7 and 8.

7.      It is better to provide the scale in Figure 8 (b).

8.      Please reorganize the Table 2.

9.      The chapter 3 should be reorganized. Some results are repeated in several figures. For example, Figure 16 is the combination of Figures 12 and 14, Figure 17 is the combination of Figures 13 and 15.

10.   The limitations of this study are suggested to be mentioned in the last section and some outlook should be mentioned.

Author Response

The authors of the Article Applsci -1890587 would like to extend thoughts of gratitude to the reviewers who took time out of their busy schedule to comment and revise this manuscript. Thank to the reviewers’ efforts, the article has been improved substantially. The authors hope that the revisions made in this version of the draft meet the requirements on the points of amendment made by the reviewers.

Reviewer 1

Comment 1

The abstract is suggested to be condensed.

Response 2

Abstract has been condensed and made clearer.

Please refer to Lines 15 to 31 in the revised manuscript.  

Comment 2

The Figure 2 is not necessary.

Response 2

Figure 2 has been removed in the revised manuscript

Comment 3

Please offer the full name for the term which is mentioned at the first time. For example, the KS.

Response 3

First mentions for full name for abbreviations have been checked and revised throughout the manuscript   

(For example in the Abstract, Line 25)

Comment 4

Some expressions in the manuscript are not clear. An English native speaker is suggested to carefully proofread it again. For example, the sentence between 108 and 112 is suggested to be reorganized into 2 shorter sentences.

Response 4

The authors have looked over the manuscript extensively with the help of a native English speaker to revise the manuscript. The authors would sincerely appreciate the time for the reviewer to once again look over the revised version of the manuscript. Extensive changes have been made to the introduction and the results section as well as other parts of the article, making sure to remove grammatical errors and incoherent syntax.

Comment 5

Some words are too small in Figure 5. It is better to use the same font size in the figures and the texts.

Response 5

Small text in Figure 5 has been revised to be bigger. Furthermore Figure 4 and 6 has been merged to one single Figure.

Comment 6

Aspect ratio in some figures seems to be changed, e.g., Figures 7 and 8.  It is better to provide the scale in Figure 8 (b).

Response 6

Upon inspection, aspect ratios for figures provided for Figure 7 and 8 are difficult to change as to not reveal details on the manufacturer trademarks. As the Figure were not critical components of the research content, the figures were removed to avoid confusion from the readers.

Comment 7

Please reorganize the Table 2.

Response 7

Table 2 has been reorganized to be clearer and easier to read (along with other tables). Please refer to Tables 2,6, 7, 8, 9 in the revised manuscript.

Comment 8

The chapter 3 should be reorganized. Some results are repeated in several figures. For example, Figure 16 is the combination of Figures 12 and 14, Figure 17 is the combination of Figures 13 and 15.

Response 8

The experimental results were maintained, but graphical results for the comprehensive comparison of the results were maintained. (Formally Figure 12~15 have been removed) Please refer to the revised manuscript for details

Comment 9

The limitations of this study are suggested to be mentioned in the last section and some outlook should be mentioned.

Response 9

Limitations of the scope of this study has been added in the conclusion section of the revised manuscript (Lines 353-358) as well as in context throughout the main body.

Reviewer 2 Report

1.       Page 2 lines 70 to 72: This whole sentence is not clear. For example, environmental hazards are a kind of environmental problem, the authors may want to specify the problems more clearly to differentiate them from the hazards. Also, “as” may not suit hereafter “use”, can replace it with “because” or “due to”.

2.       Page 2 line 72: are there any downsides of the modified asphalt sheet to be used to prevent leakage? The authors need to clarify that of fulfilling research objectives.

3.       Page 3 line 86: a typo

4.       Table 1: what the unit? How do authors define adhesive performance here?

5.       Table 5: The capital letters in English in the table need a note for explanation. By the way, what is W/B ratio? Do authors mean W/C ratio here (water-cementitious ratio)?

6.       Figure 9. Legend is wrong. (c) is LMC and (d) is APC.

7.       In table 6 and 7, use “>” rather than “more than”.

8.       Table 7. Explanations are needed for the sample treatments.

9.       Literature review is needed to summarize similar work. There are lots of papers and products about waterproof membranes so authors need to emphasize what is unique from their own products. Also, the cost is the main factor. How much the cost has gone up for your modified sheet needs to be listed as well. 

Author Response

The authors of the Article Applsci -1890587 would like to extend thoughts of gratitude to the reviewers who took time out of their busy schedule to comment and revise this manuscript. Thank to the reviewers’ efforts, the article has been improved substantially. The authors hope that the revisions made in this version of the draft meet the requirements on the points of amendment made by the reviewers.

Reviewer 2

Comment 1

   Page 2 lines 70 to 72: This whole sentence is not clear. For example, environmental hazards are a kind of environmental problem, the authors may want to specify the problems more clearly to differentiate them from the hazards. Also, “as” may not suit hereafter “use”, can replace it with “because” or “due to”.

Response 1

The authors have looked over the manuscript extensively with the help of a native English speaker to revise the manuscript. The authors would sincerely appreciate the time for the reviewer to once again look over the revised version of the manuscript. Extensive changes have been made to the introduction and the results section as well as other parts of the article, making sure to remove grammatical errors, colloquial expressions and generic statements.

Comment 2

Page 2 line 72: are there any downsides of the modified asphalt sheet to be used to prevent leakage? The authors need to clarify that of fulfilling research objectives.

Response 2

Please refer to the revised manuscript Lines 66 to 78. Conventional waterproofing sheets are more easily susceptible to shear stress and are not commonly used. Membrane coating types are preferred due to how easy it is to install as well as low cost considerations. Multilayered waterproofing sheets that have double-sided bonding are more difficult to install and are more costly, but have reportedly been able to provide successful waterproofing performance in the long term (Incheon International Airport subway system is an example often used in Korea).

Comment 3

 Page 3 line 86: a typo

Response 3

Revised. Please refer to the revised manuscript Line 79.

Comment 4

Table 1: what the unit? How do authors define adhesive performance here?

Response 4

Specific adhesion strength values have been reported in the revised Table 1.

Comment 5

Table 5: The capital letters in English in the table need a note for explanation. By the way, what is W/B ratio? Do authors mean W/C ratio here (water-cementitious ratio)?

Response 6

Abbreviations for Table 5 have been clarified and revised

Comment 6

Figure 9. Legend is wrong. (c) is LMC and (d) is APC.

Response 6

Revised. Thank you kindly for the point of revision.

Comment 7

In table 6 and 7, use “>” rather than “more than”.

Response 7

Revised in Table 6, 7 and other relevant tables

Comment 8

Table 7. Explanations are needed for the sample treatments.

Response 8

Table 7 has been reorganized such that performance requirements and results are expressed more clearly

Comment 9

Literature review is needed to summarize similar work. There are lots of papers and products about

waterproof membranes so authors need to emphasize what is unique from their own products. Also, the cost is the main factor. How much the cost has gone up for your modified sheet needs to be listed as well. 

Response 9

Literature review has been included in the introduction section (lines 47-59), but it must be noted that most existing studies are centered on either removal of waterproofing layer by improving the properties of the pavement layer itself (which is the result of the LMC pavement), and/or improving the adhesion performance of the existing asphalt overlay on concrete pavement. There aren’t too many papers that discuss the prospects of replacing the asphalt system in lieu of the existing asphalt overlay/ waterproofing layer that can also be applied to pavement systems that are designed to not use waterproofing. This is a common engineering issue, particularly in countries such as Korea and China where highspeed bridges are being constructed in recent times, where cost efficiency is prioritized over quality and long-term performance. Despite extensive research and application of new technologies, it has been undisputed that bridges with waterproofing require less frequent maintenance than those without, but are obviously far higher in cost during construction. As this proposed new waterproofing material is currently an item intended for patent application, there is no readily available data on the projected costs analysis, but we expect that in terms of initial construction costs, it will be much higher. However, significant advantages of the cost benefits for maintenance and repair is also expected. The authors want to emphasize that the cost analysis on initial costs vs long-term cost savings may deter the readers from missing the crucial underlying component of the article that regardless of the pavement system, it is highly recommended to use a waterproofing layer to ensure long life cycle of the bridge structures, and thus we are hopeful that discussing primarily on the performance evaluation/comparison suffices for the goal of this article.    

Reviewer 3 Report

Dear Sirs,

First of all in the manuscript no division of chapters into research results and discussion and lack of clear stated research parametres. In the research assumptions, the authors forgot why on the bridge decks waterproofing layers made of reinforced bitumen sheets or  reinforced modified acrilic emulsion are not used with concrete pavements. Namely due to the different thermal expansion of such products under operating conditions, causing mechaniczal damage to both products working in the direct contact of these layers. Such basic problem was not taken into account in the evaluation of the results. For this reason , it seems the rong conclusion " in this study is expected to show stable waterproof performance even when applied to the LMC pavement"

The presentation of the research results refers to the Korean national standards, witout specifying the research parameters, which is incomprehensible to readers from other countries. Research descripions need to be suplemented. Entries in tables and in some figures are incomprehensible, e.g:

- in table 1- no explonation as to how to specify digital values (units?),

- the captions to figures 7 and 8 are incomprehensible . It should be brriefly discussed what the authors want to show in these figures,- the korean standard in scetion 2.1.3, 2.3.1, 2.3.2 is unknown to other readers ouside Korea. The test method for specified properties shall be shortly described,

- the entries in tables 2,6, 7, 8, 9 are incomprehensible = provide this information in a legible manner,

- lack of information on the test conditions makes it impossible to understand the provisions regarding the evaluation of the results in poin 3.1.1

- reference no10 i.e. "Code of practice for protection of below ground structures against water from the  ground,” British Standards Institution, seems to be not wright to such statement" ....….the modified asphalt sheet has been mainly used  in Korean market"

with kind regards

Author Response

The authors of the Article Applsci -1890587 would like to extend thoughts of gratitude to the reviewers who took time out of their busy schedule to comment and revise this manuscript. Thank to the reviewers’ efforts, the article has been improved substantially. The authors hope that the revisions made in this version of the draft meet the requirements on the points of amendment made by the reviewers.

Reviewer 3

Comment 1

In my opinion, the authors should delete the first sentence of the first paragraph (lines 40-41), it is not a technical phrase, and it is too generic statement.

Comment 2

Authors should thoroughly review expressions in English such as: "And as the application of sustainable high performance waterproofing technology is required (...)" in line 50-51, and the colloquial expressions as: “If we look at the current bridge (…)” or “In other words (…)” in line 56.

Response 1 and 2

The authors have looked over the manuscript extensively with the help of a native English speaker to revise the manuscript. The authors would sincerely appreciate the time for the reviewer to once again look over the revised version of the manuscript. Extensive changes have been made to the introduction and the results section as well as other parts of the article, making sure to remove grammatical errors, colloquial expressions and generic statements.

Comment 3

the introductory section must include a review of the state of the art, and in the first paragraphs there are few references included.

Response 3

Literature review has been included in the introduction section (lines 47-59), but it must be noted that most existing studies are centered on either removal of waterproofing layer by improving the properties of the pavement layer itself (which is the result of the LMC pavement), and/or improving the adhesion performance of the existing asphalt overlay on concrete pavement. There aren’t too many papers that discuss the prospects of replacing the asphalt system in lieu of the existing asphalt overlay/ waterproofing layer that can also be applied to pavement systems that are designed to not use waterproofing. This is a common engineering issue, particularly in countries such as Korea and China where highspeed bridges are being constructed in recent times, where cost efficiency is prioritized over quality and long-term performance. Despite extensive research and application of new technologies, it has been undisputed that bridges with waterproofing require less frequent maintenance than those without, but are obviously far higher in cost during construction. As this proposed new waterproofing material is currently an item intended for patent application, there is no readily available data on the projected costs analysis, but we expect that in terms of initial construction costs, it will be much higher. However, significant advantages of the cost benefits for maintenance and repair is also expected. The authors want to emphasize that the cost analysis on initial costs vs long-term cost savings may deter the readers from missing the crucial underlying component of the article that regardless of the pavement system, it is highly recommended to use a waterproofing layer to ensure long life cycle of the bridge structures, and thus we are hopeful that discussing primarily on the performance evaluation/comparison suffices for the goal of this article.   

Comment 4

I suggest that in Figure 4 not only the existing types of pavements based on conventional waterproofing method be included, but also a scheme of the proposed waterproofing sheet that can be applied to both APC and LMC. Perhaps figure 4 and figure 5 could be merged into a single figure. Figure 4, 5 and 6 can be reduced to one or two figures. Similar graphic information is included in these three Figures.

Response 4

Figure 4 and 6 has been merged to one single Figure,

Comment 5

The paper in focused on (1) the adhesion characteristics of waterproofing layer and APC through withstanding the high temperature and (2) rolling pressure were evaluated. However, other properties could also have been studied, such as the environmental evaluation of performing this modification to the composition of pavement.

Response 5

As has been aptly pointed out by the reviewer, other properties could have been studied, but the effect of environmental considerations as far as the entirety of the pavement structure on a bridge infrastructure (or any road pavement) is concerned, environmental factors directly affect the road pavement, and not the waterproofing layer that is installed underneath the pavement overlay. That being said, KS testing method used in this study (KS F 4931), which has been drafted in accordance to similar methods found in ASTM and JS, investigates the environmental response performance of the waterproofing material. The results of this basic property investigation is outlines in Table 7.

However, the core investigation concerned in this study (shear and tensile adhesive strength and shear deformation) is the priority which is outline from Section 3.2 and onwards. This is because the primary concern regarding waterproofing (overlay or sheet) and pavement is the problem with homogenous integration, which is adhesion. As is commonly known, asphalt overlay and asphalt/concrete pavement is very commonly known to have multitudes of defects related to adhesion issues. The same can be said about waterproof coating and sheets with concrete and asphalt pavement. How well the waterproofing material integrates with the pavement is the core, before we investigate the environmental effects on the waterproofing layer.

Comment 6

Table 2 about the physical property evaluation criteria and performance standard of waterproofing sheet is too complex. Please, improve the way information of properties is presented. In my opinion, (as Table 2) the way that information of properties is presented must be improved.

Response 6

 Table 2 has been reorganized to be clearer and easier to read (along with other tables). Please refer to Tables 2,6, 7, 8, 9 in the revised manuscript.

Comment 7

Sections 2.2.1, 2.2.2, 2.2.3 are clearly described, but possibly it would have been more appropriate to group the three short sections into a single section.

Response 7

Subsections 2.2.1, 2.2.2, and 2.2.3 have been merged into a single section. Please refer to the Revised Section 2.2 from Line 169 for details

Comment 8

Although this section makes a complete discussion of the results obtained, and the graphic results are clear, it is missing contrasting the main data obtained in the laboratory tests with previous experiences of other researchers. It is essential to compare our own results with previous experiments, this would reinforce the findings.

Response 8

While the reviewer has pointed out a key concern with this article, the authors would like to report to the reviewer that there are no similar comparable experiments that has been conducted in other fields of research regarding double-sided bonded waterproofing sheets applicable with bridge (or road) pavement structures. There ARE researches on waterproofing membranes, coating and overlay/protection layers that can be referred to such as one conducted by Tariq Usman Saeed et al., Effects of Bridge Surface and Pavement Maintenance Activities on Asset Rating”:, and by Jiancun Fu “Study on the Influence and Law of Waterproof System Design Factors on the Typical Stress of Bridge Deck Pavement.” (there are too many to list). However, these research are not comparable as the focus is on emphasizing the importance and/or necessity of conventional waterproofing technology. The focus of this paper is centered on double-sided bonding waterproof sheet, which is not a conventional waterproofing material for bridge or road infrastructures, therefore comparison with previous experiences by existing researcher will undoubtedly come at odds.

References to other researcher findings has been included however, in lines 48 to 59 in the revised manuscript.

Comment 9

The main conclusion is that the double-sided bonding composite waterproofing sheet proposed as a new waterproofing material, presented an adequate technical behavior and its feasibility was proved. For that, I recommend to include the word “feasible” instead of “verified.” A general conclusion is necessary, and the 4 bullet points must be slightly summarized, in my opinion, although the conclusion section is adequate and clear, it is excessively long.

Response 9

Conclusion section has been revised to reflect on the possibility of using this proposed material as an option when reinforced waterproofing performance is required for the bridge structure in Lines 331 – 340 in the revised manuscript (as well as in the main body). Please refer to the revised manuscript for details

Reviewer 4 Report

The manuscript entitled “Property Analysis of Double-Sided Composite Waterproofing 2 Sheet for Simultaneous Application on Asphalt Concrete and 3 Latex Modified Concrete Pavements for Bridge Decks” presents a solid development of analysis based on ensure long term durability and safety of bridge deck slabs.

The proposal of a double-side bonding composite waterproof is interesting to be installed in both APC pavement and LMC.

However, the following comments are formulated after the review process:

Introduction section:

In my opinion, the authors should delete the first sentence of the first paragraph (lines 40-41), it is not a technical phrase, and it is too generic statement.

Authors should thoroughly review expressions in English such as: "And as the application of sustainable high performance waterproofing technology is required (...)" in line 50-51, and the colloquial expressions as: “If we look at the current bridge (…)” or “In other words (…)” in line 56.

It is recommended that the manuscript be reviewed by an English editing service deeply.

the introductory section must include a review of the state of the art, and in the first paragraphs there are few references included.

I suggest that in Figure 4 not only the existing types of pavements based on conventional waterproofing method be included, but also a scheme of the proposed waterproofing sheet that can be applied to both APC and LMC. Perhaps figure 4 and figure 5 could be merged into a single figure.

Section 2. Properties and test methods of newly proposed waterproofing sheet and layer:

The paper in focused on (1) the adhesion characteristics of waterproofing layer and APC through withstanding the high temperature and (2) rolling pressure were evaluated. However, other properties could also have been studied, such as the environmental evaluation of performing this modification to the composition of pavement.

Figure 4, 5 and 6 can be reduced to one or two figures. Similar graphic information is included in these three Figures.

Table 2 about the physical property evaluation criteria and performance standard of waterproofing sheet is too complex. Please, improve the way information of properties is presented.

Section 2.2. Preparation of pavement and deck concrete materials:

Sections 2.2.1, 2.2.2, 2.2.3 are clearly described, but possibly it would have been more appropriate to group the three short sections into a single section.

Section 2.3.1. Specimen preparation and test methods for performance evaluation of waterproofing sheet:

This section is clearly written. The graphical information included is really helpful to understand the laboratory experimental sequence.

Table 6. Evaluation criteria and quality standards of waterproofing sheet:

In my opinion, (as Table 2) the way that information of properties is presented must be improved.

Section 3. Evaluation results and considerations:

Although this section makes a complete discussion of the results obtained, and the graphic results are clear, it is missing contrasting the main data obtained in the laboratory tests with previous experiences of other researchers.

It is essential to compare our own results with previous experiments, this would reinforce the findings.

5. Conclusión:

The main conclusion is that the double-sided bonding composite waterproofing sheet proposed as a new waterproofing material, presented an adequate technical behaviour and its feasibility was proved. For that, I recommend to include the word “feasible” instead of “verified”

A general conclusion is necessary, and the 4 bullet points must be slightly summarized, in my opinion, although the conclusion section is adequate and clear, it is excessively long.

Author Response

The authors of the Article Applsci -1890587 would like to extend thoughts of gratitude to the reviewers who took time out of their busy schedule to comment and revise this manuscript. Thank to the reviewers’ efforts, the article has been improved substantially. The authors hope that the revisions made in this version of the draft meet the requirements on the points of amendment made by the reviewers.

Reviewer 4

Comment 1

First of all, in the manuscript no division of chapters into research results and discussion and lack of clear stated research parameters.

Response 1

Please refer to the revised version of the manuscript. The authors have looked over the manuscript extensively to revise the manuscript and extensive changes have been made to the introduction and the results section as well as other parts of the article, making sure to remove grammatical errors, colloquial expressions and generic statements, and clarify the research parameters.

Comment 2

In the research assumptions, the authors forgot why on the bridge decks waterproofing layers made of reinforced bitumen sheets or\ reinforced modified acrylic emulsion are not used with concrete pavements. Namely due to the different thermal expansion of such products under operating conditions, causing mechanical damage to both products working in the direct contact of these layers. Such basic problem was not taken into account in the evaluation of the results. For this reason, it seems the wrong conclusion " in this study is expected to show stable waterproof performance even when applied to the LMC pavement"

Response 2

This issue is addressed in the newly revised manuscript, lines 61 – 95. As has been aptly pointed out by the reviewer, concrete pavements (LMC included) are designed without a separate layer of waterproofing material installation in mind. While this does not always pose a problem with every concrete pavement based bridge structures, it must still be noted that pavements will be subject to cracks. In the case of bridges, this a commonly occurring problem (and very apparent in East Asia including Vietnam, Korea, China and Singapore) where leakage led to degradation of bridges and roads, and often leads to increased maintenance costs. Therefore, even when construction determines to use LMC, the option to employ a suitable waterproofing technology should be made available, and the proposed waterproofing sheet investigated in this study is intended to serve as this option.

The proposed waterproofing sheet in this study (while field verification still is required of course, as well as applicability in other countries) has two distinct properties that is supposed to overcome the outlined concerns related to waterproofing with concrete pavements 1) it has double sided bonding with high adhesion strength such that deflections and shear deformation will not cause adhesion failure, and 2) the asphalt membrane component has high resistance to crack displacement that will eventually occur on the concrete pavement due to its viscoelastic property.

Field application in the Korean research report has already shown ample results in providing stable waterproofing performance, but understanding the concerns from the reviewer, we have revised the conclusion in the article to state that “further investigation will be required before the proposed double-bonding waterproofing sheet is feasible to provide stable waterproof performance even when applied to LMC pavement.”

Comment 3

The presentation of the research results refers to the Korean national standards, without specifying the research parameters, which is incomprehensible to readers from other countries. Research descriptions need to be supplemented.

Response 3

The following lines have been included  in Sections 2.1.3, and , 2.3.2

“(methodology of the parameters outlined in KS F 4931 (Table 2) are in compliance with similar methods found in the American Society of Testing Methods (ASTM) as KS F 4931 was drafted using ASTM and Japanese Standard (JS) as reference)”

Comment 4

Entries in tables and in some figures are incomprehensible, e.g:- in table 1- no explanation as to how to specify digital values (units?),

Response 4

Specific adhesion strength values have been reported in the revised Table 1.

Comment 5

- the captions to figures 7 and 8 are incomprehensible. It should be briefly discussed what the authors want to show in these figures,

Response 5

Upon inspection, aspect ratios for figures provided for Figure 7 and 8 are difficult to change as to not reveal details on the manufacturer trademarks. As the Figure were not critical components of the research content, the figures were removed to avoid confusion from the readers.

Comment 6

- the Korean standard in section 2.1.3, 2.3.1, 2.3.2 is unknown to other readers outside Korea. The test method for specified properties shall be shortly described,

The following lines have been included  in Sections 2.1.3, and , 2.3.2

(methodology of the parameters outlined in KS F 4931 (Table 2) are in compliance with similar methods found in the American Society of Testing Methods (ASTM) as KS F 4931 was drafted using ASTM and Japanese Standard (JS) as reference)

Comment 7

- the entries in tables 2,6, 7, 8, 9 are incomprehensible = provide this information in a legible manner,

Response 7

 Tables has been reorganized to be clearer and easier to read (along with other tables). Please refer to Tables 2,6, 7, 8, 9 in the revised manuscript.

Comment 8

- lack of information on the test conditions makes it impossible to understand the provisions regarding the evaluation of the results in point 3.1.1

Response 8.

Evaluation results in Point 3.1.1 is meant to be results on the basic physical properties of the waterproofing material to ensure that the waterproofing material passes as a satisfactory waterproofing material. Explanations on the context of this testing method and results has been included in lines 156 to 166 in the revised manuscript.  

Comment 9

- reference no10 i.e. "Code of practice for protection of below ground structures against water from the  ground,” British Standards Institution, seems to be not wright to such statement" ....….the modified asphalt sheet has been mainly used  in Korean market"

Response 9

Reference 10 has been removed as this was a numbering mistake made by the authors

Round 2

Reviewer 2 Report

1. Table 1. What's the definition of "ratio"? Is it by weight or volume? The numbers shown in the Table, are they percentages? Among these components, which one has more impact on adhesive performance?

2. Line 193: specimen fabrication. What's the specimen curing temperature and humidity? What are the dimensions of your concrete specimens?

3. Tables 3 and 4: again, are ratios based on weight or volume?

4. Figure 17: how many replicates were completed? 

Author Response

Comment 1

Table 1. What's the definition of "ratio"? Is it by weight or volume? The numbers shown in the Table, are they percentages? Among these components, which one has more impact on adhesive performance?

Response 1

Table 1 and explanations on Table 1 has been revised to specify the details on the ratio. Numbers are regarding weight ratio percentage. The definition of ratio is based on the weight percentage (Lines 142-150 amended to clarify this point). As the modified latex acrylic emulsion is part of a product that is currently under the process of patent application, it is not within the permission of the authors to disclose the details on the mixture ratios, but the ratio of additives and new synthetic resin is the key determining factor for the adhesion performance of the emulsion. Lines 154 to 161 in the revised manuscript address this concern.

Comment 2

Line 193: specimen fabrication. What's the specimen curing temperature and humidity? What are the dimensions of your concrete specimens?

Response 2

The following has been added/revised in the manuscript in Lines 197 to 205;

The specimens of APC and LMC pavement applied waterproofing sheet specimens were fabricated according to the specimen manufacturing method of the Korean Standard KS F 4931-“17” 『Sheet Waterproofing Materials for Concrete Bridges』as shown in Figure 5, and placed in the constant room temperature setting (20℃±3, RH 65%) for curing for 3 days. 20 × 20 cm sized double-sided composite waterproof sheet are applied for each specimen type, and thickness of the concrete substrate (bottom part) and the pavement substrate (upper part) are prepared. For each substrate type (APC and LMC) three specimens were prepared.  

Comment 3

Tables 3 and 4: again, are ratios based on weight or volume?

Response 3

Tables 3 and 4 and explanations on Table 3 and 4 has been revised to specify the details on the ratio. Numbers are regarding weight ratio percentage. (Lines 181 to 183)

Comment 4

Figure 17: how many replicates were completed? 

Response 4

Results outlined in Figure 17 (Revised to Figure 9 in the manuscript) are based on 3 repeated specimens. This has been clarified in lines 312 in the revised manuscript.